# The Effects of Drying Techniques on Phytochemical Contents and Biological Activities on Selected Bamboo Leaves

**DOI:** 10.3390/molecules27196458

**Published:** 2022-09-30

**Authors:** Mohammad Amil Zulhilmi Benjamin, Shean Yeaw Ng, Fiffy Hanisdah Saikim, Nor Azizun Rusdi

**Affiliations:** Institute for Tropical Biology and Conservation, Universiti Malaysia Sabah, Jalan UMS, Kota Kinabalu 88400, Sabah, Malaysia

**Keywords:** drying methods, antioxidant activities, brine shrimp lethality assay, phytochemical contents, bamboo leaves

## Abstract

The therapeutic potential of bamboos has acquired global attention. Nonetheless, the biological activities of the plants are rarely considered due to limited available references in Sabah, Malaysia. Furthermore, the drying technique could significantly affect the retention and degradation of nutrients in bamboos. Consequently, the current study investigated five drying methods, namely, sun, shade, microwave, oven, and freeze-drying, of the leaves of six bamboo species, *Bambusa multiplex*, *Bambusa tuldoides*, *Bambusa vulgaris*, *Dinochloa sublaevigata*, *Gigantochloa levis*, and *Schizostachyum brachycladum*. The infused bamboo leaves extracts were analysed for their total phenolic content (TPC) and total flavonoid content (TFC). The antioxidant activities of the samples were determined via the 2,2-diphenyl-1-picrylhydrazyl (DPPH), 2,2′-azino-bis(3-ethylbenzothiazoline-6-sulfonic acid) (ABTS), and ferric reducing antioxidant power (FRAP) assays, whereas their toxicities were evaluated through the brine shrimp lethality assay (BSLA). The chemical constituents of the samples were determined using liquid chromatography–tandem mass spectrometry (LC-MS/MS). The freeze-drying method exhibited the highest phytochemical contents and antioxidant activity yield, excluding the *B. vulgaris* sample, in which the microwave-dried sample recorded the most antioxidant and phytochemical levels. The TPC and TFC results were within the 2.69 ± 0.01–12.59 ± 0.09 mg gallic acid equivalent (GAE)/g and 0.77 ± 0.01–2.12 ± 0.01 mg quercetin equivalent (QE)/g ranges, respectively. The DPPH and ABTS IC_50_ (half-maximal inhibitory concentration) were 2.92 ± 0.01–4.73 ± 0.02 and 1.89–0.01 to 3.47 ± 0.00 µg/mL, respectively, indicating high radical scavenging activities. The FRAP values differed significantly between the drying methods, within the 6.40 ± 0.12–36.65 ± 0.09 mg Trolox equivalent (TE)/g range. The phytochemical contents and antioxidant capacities exhibited a moderate correlation, revealing that the TPC and TFC were slightly responsible for the antioxidant activities. The toxicity assessment of the bamboo extracts in the current study demonstrated no toxicity against the BSLA based on the LC_50_ (lethal concentration 50) analysis at >1000 µg/mL. LC-MS analysis showed that alkaloid and pharmaceutical compounds influence antioxidant activities, as found in previous studies. The acquired information might aid in the development of bamboo leaves as functional food items, such as bamboo tea. They could also be investigated for their medicinal ingredients that can be used in the discovery of potential drugs.

## 1. Introduction

Drying is a crucial stage during post-harvest because it aids in preventing enzymatic breakdown and microbial development while retaining the beneficial characteristics of the dried plants [1]. Plant leaves are dried either naturally or via artificial methods. Conventional techniques, such as open sun- and shade-drying at ambient temperatures, are still employed in rural regions [1]. Nevertheless, due to the uncontrollable conditions of the methods, guaranteeing the safety, efficacy, and consistency of the dried products represents a challenge [2]. Currently, conventional air oven-drying is a standard technique to dry food, although it frequently alters the nutritional value, flavour, and texture of the food and might oxidise and degrade heat-sensitive polyphenols [3,4]. Alternatively, various artificial drying methods, including microwave and freeze-drying, have been utilised to rapidly dry substantial amounts of leaves with adequate quality [5]. Hence, fresh bamboo has a high moisture content and needs drying to avoid microbial damage and mould, making it ready for further processing, storage, transportation, and utilisation [6].

The nutritive and therapeutic potential of bamboo leaf extracts in the food and pharmaceutical industries have garnered attention worldwide [7]. Biologically, bamboo leaves are rich in polyphenols, flavonoids, and other secondary plant metabolites [8,9]. The plants are also widely utilised in traditional Asian medicine to treat arteriosclerosis, cardiovascular disease, hypertension, certain cancers, oedema, diarrhoea, vomiting, extreme thirst, and to improve the flavour and colour of foods [8,9]. Moreover, bamboos are natural antioxidants that possess the potential to be utilised as novel food additives in edible oils, fish, and meat products [10].

For many years, bamboo leaf tea has been considered a delicious and healthy drink in Asian countries [11]. Some edible bamboos, such as *Bambusa* sp., are consumed as tea and pickles due to their high nutritional and mineral values [12]. Zhucha, an ancient Uyghur treatment, is produced from bamboo leaves and green tea, which possess superior effectiveness and lipid-reducing effects [13]. *Sasa quelpaertensis*, a bamboo species endemic to Jeju Island, South Korea, is ingested as a medicinal tea for its anti-diabetic, diuretic, and anti-inflammatory properties [14]. In Japan, *Sasa veitchii* (or Kuma-zasa) is widely employed as an ornamental food trimming and in folk medicines. Furthermore, Kuma-zasa leaves have been utilised in the medical field to treat burns and urinary hesitancy [15].

Phenolics and flavonoids constituents are responsible for the functional efficacy of herbs. Drying herbal plants could inhibit bacterial growth, increase sample quality, and prevent the oxidation of their chemical contents. Consequently, the drying technique employed could considerably affect the degradation of the phytochemical and antioxidant contents of a plant. In this regard, toxicity studies should be accommodated in parallel with antioxidant activity to ensure their safe use as functional food materials. Nevertheless, the impacts of the methods on the quality of bamboo leaves have not been explored in depth. The present study investigated bamboo leaves and optimised their appropriate drying processes by evaluating the effects of five different drying methods (sun, shade, microwave, oven, and freeze-drying) on the phytochemical content and antioxidant activities of different bamboo species. Moreover, the toxicity effects were determined through a brine shrimp lethality assay (BSLA) of the six bamboo species selected. Using liquid chromatography–tandem mass spectrometry (LC-MS/MS), further investigation was performed on the chemical constituents of six different types of bamboo species.

## 2. Results and Discussion

### 2.1. The Phytochemical Contents

#### 2.1.1. Total Phenolic Content

The total phenolic content (TPC) data of the six bamboo species assessed in the current study are presented in Table 1. Drying considerably increased the TPC (*p* < 0.05) of the extracts, and the increment pattern was the lowest in the sun-dried *G. levis*, followed by *S. brachycladum*, *B. vulgaris*, *B. tuldoides*, *B. multiplex*, and *D. sublaevigata*. Similar results were documented by Singhal et al. [16], who reported that sun-dried *B. vulgaris* shoots recorded the lowest TPC (195.05 ± 9.82) compared with the tray- (229.6 ± 54.25), oven- (227.55 ± 7.77), microwave- (224.95 ± 49.05), and freeze- (227.66 ± 87.12) dried specimens. The long drying time, which exposed the samples to the atmosphere, resulted in degradation from the oxidation of the phenolic compounds and might explain the low TPC of the sun-dried samples [17]. Enzymatic reactions might also contribute to the loss of the phenolic chemicals during conventional drying procedures [16].

The drying techniques employed in the present study were incapable of inactivating degradative enzymes, such as polyphenol oxidases, which were responsible for degrading phenolic compounds during lengthy drying periods [18]. The stability of phenolic chemicals in herbal infusions was also reported to be affected by drying temperatures [19]. The results demonstrated that the freeze-dried *B. multiplex*, *D. sublaevigata*, *G. levis*, and *S. brachycladum* recorded the highest TPC. The findings aligned with a report that recorded low-temperature drying techniques, including freeze-, vacuum-, and infrared-radiation-drying, enhanced the retention of bioactive chemicals and antioxidant activities in *Dendrobium officinale* bamboo shoots [20].

The influences of harvesting season and drying method on the phenolics, flavonoids, triterpenoids, and antioxidative activities of the leaves of two bamboo species, *Pleioblastus kongosanensis* f. *aureostriatus* and *Shibatea chinensis*, were observed [21]. The study also found that freeze-, vacuum-, and microwave-oven-drying procedures resulted in significantly different outcomes [21]. In another investigation, microwave-drying techniques produced the highest quality dried herbs faster than other methods [22]. Similarly, the *B. tuldoides* and *B. vulgaris* specimens evaluated in this study subjected to microwave-drying documented the highest TPC.

#### 2.1.2. Total Flavonoid Content

Table 1 summarises the total flavonoid content (TFC) of the bamboo samples assessed in the current study. Drying notably (*p* < 0.05) enhanced the TFC of the bamboo extracts. The sun-dried *B. tuldoides*, *B. vulgaris*, *G. levis*, and *S. brachycladum* specimens recorded the lowest TFC. Although the oven-dried *B. multiplex* and shade-dried *D. sublaevigata* specimens documented the least TFC, most reports suggested that sun-drying might not be a satisfactory technique for some herbs [22]. Although the plants are placed in the shade out of direct sunlight, shade-drying also utilises solar energy as the heat source, which is similar to sun-drying. Nonetheless, the approach is disadvantageous because it requires unusually extended drying durations [23], which could promote the development of insects and moulds in high relative air humidity [24].

The freeze-dried *B. multiplex*, *B. tuldoides*, *D. sublaevigata*, *G. levis*, and *S. brachycladum* specimens in this study demonstrated the highest TFC, supporting the report by Soesanto [25], who noted that the freeze-dried extracts of *B. vulgaris* and *G. apus* shoots exhibited higher TFC, TPC, and DPPH than the oven-dried samples. Nonetheless, the microwave-dried *B. vulgaris* in this study recorded the highest TFC at 0.14 mg gallic acid equivalent (GAE)/g, slightly dissimilar from the freeze-dried extracts that exhibited the second highest TFC. Singhal et al. [16] reported that *B. vulgaris* shoots recorded the second highest TFC yield when microwave-dried (371.24 ± 17.24) following freeze-drying (438.29 ± 6.39), which were superior compared with tray- (284.87 ± 34.95), sun- (346.86 ± 26.15), and oven- (327.01 ± 19.19) drying.

### 2.2. Antioxidant Activities

#### 2.2.1. The 2,2-diphenyl-1-picrylhydrazyl Assay

The 2,2-diphenyl-1-picrylhydrazyl (DPPH) assay is commonly employed to evaluate the antioxidant activities of samples because the method is simple and inexpensive, and requires little operating skill and a simple spectrophotometer [26]. The IC_50_ (half-maximal inhibitory concentration) value is widely utilised to assess the antioxidant activities of the samples and is determined from the concentration of antioxidants required to diminish the initial DPPH concentration by 50% [27]. A smaller IC_50_ value indicates better antioxidant attributes.

The DPPH results of this study are presented in Table 2. The sun-dried bamboo samples recorded high DPPH IC_50_ values (lowest free radical scavenging activity). At the same time, the freeze-dried *B. multiplex*, *B. tuldoides*, *D. sublaevigata*, *G. levis*, and *S. brachycladum* and microwave-dried *B. vulgaris* extracts exhibited superior DPPH free radical scavenging activities. One report suggested that a diminished antioxidant content was mainly attributed to oxidation processes or thermal degradation [28]. Consequently, several investigations have suggested that freeze-drying is preferable in conserving antioxidants [28,29].

Freeze-dried aqueous leaf extracts of *G. levis*, *G. scortechinii*, and *S. zollingeri* exhibited a higher DPPH yield than ethanolic extracts [30]. *Fargesia robusta* (clumping bamboo) also demonstrated the highest antioxidant capacity for DPPH when its aqueous methanolic leaf extract was freeze-dried [31]. In another study, Kozlowska et al. [32] noted that the antiradical properties of freeze-dried herbal materials (coriander, tarragon, lovage, and Indian borage) possessed considerably (*p* < 0.05) superior DPPH scavenging abilities than the fresh raw material. Nonetheless, the microwave-dried *B. vulgaris* sample in this study was subjected to higher microwave power or temperature, thus resulting in increased antioxidant activity and TPC [33,34].

#### 2.2.2. The 2,2′-azino-bis(3-ethylbenzothiazoline-6-sulfonic acid) Assay

In addition to DPPH, 2,2′-azino-bis(3-ethylbenzothiazoline-6-sulfonic acid) (ABTS) is one of the most commonly employed assessments to evaluate the antioxidant activities of plant extracts, foods, and unique compounds [35]. The obtained IC_50_ value reflects the antioxidant activities of test samples because it records the concentration required to produce 50% inhibition. Accordingly, the lower the IC_50_ value, the higher the antioxidant activity.

Table 2 lists the ABTS values of the bamboo samples evaluated in the current study. The samples dried under the sun yielded the lowest antioxidant activity, excluding the fresh sample. The IC_50_ values of the *B. multiplex*, *B. tuldoides*, *B. vulgaris*, *D. sublaevigata*, and *G. levis* samples were the most significant, whereas the shade-dried *S. brachycladum* contributed the highest IC_50_ value. Saifullah et al. [36] reported that the antioxidant capacity measured via ABTS of the sun- and shade-dried lemon myrtle exhibited the lowest yield compared with hot-air-, vacuum-, and freeze-dried specimens.

The highest antioxidant activities or the lowest IC_50_ values were recorded by the oven-dried *B. multiplex*, *B. tuldoides*, *B. vulgaris*, *D. sublaevigata*, and *S. brachycladum*. Nevertheless, only the *G. levis* sample exhibited the least IC_50_ value when freeze-dried. Chuyen et al. [37] documented that the peels of Gac fruits which were hot-air-dried at 60 and 80 °C and vacuum-dried at 50 °C exhibited the highest ABTS antioxidant capacities. Consequently, hot-air- and vacuum-drying at 80 and 50 °C were recommended for drying Gac peel. Hot-air-, vacuum-, and freeze-drying methods were recommended to preserve the ABTS antioxidant activity of lemon myrtle because these techniques produced significant values compared with other methods [36]. Hot-air-drying at 40–60 °C was recommended for herbs [38], which explained the highest antioxidant activity in ABTS of the oven-dried samples in this study that was conducted at 50 °C.

#### 2.2.3. The Ferric Reducing Antioxidant Power Assay

The reducing properties of the samples in this study were assessed via the ferric reducing antioxidant power (FRAP) assay. The results varied significantly (*p* < 0.05) between treatments applied (see Table 2). The sun-dried *B. multiplex*, *B. tuldoides*, *B. vulgaris*, *D. sublaevigata*, and *G. levis* samples, and shade-dried *S. brachycladum* samples, exhibited considerably reduced FRAP contents. The results aligned with one report which demonstrated that traditionally dried spearmints, particularly sun- and shade-dried extracts, demonstrated a notably diminished FRAP compared with freeze-dried samples [39]. Traditional drying methods, such as sun- and shade-drying, possess numerous drawbacks due to their inability to produce the high-quality standards required for medicinal plants [24].

Freeze-dried aqueous extracts of *B. multiplex*, *D. sublaevigata*, *G. levis*, and *S. brachycladum* resulted in the highest FRAP amount. Kong et al. [40] noted that the reducing power of the *Clinacanthus nutans* leaf extract when freeze-dried rose with increased value (mg TE/g), where the highest level was noted in the aqueous extract (10.07 ± 0.10), followed by the ethanolic (8.34 ± 0.14) and acetone (3.24 ± 0.30) samples. Nevertheless, the microwave-dried *B. tuldoides* and *B. vulgaris* specimens in the current study recorded the highest FRAP yield. Similarly, Lasano et al. [41] recommended microwaving fermented and unfermented *Strobilanthes crispus* tea to obtain preferable antioxidant capacities, including FRAP and DPPH. The antioxidant index proposed by the report might also be employed as a new marker in determining the optimal drying method for varying food products [41].

### 2.3. The Correlation between Phytochemical Contents and Antioxidant Capacities

The antioxidant attributes of a plant extract are often associated with its polyphenols content; therefore, the correlation between its phytochemical contents and the antioxidant capacity of the selected bamboo extracts in this study dried with different methods was analysed, and the outcomes are summarised in Table 3. The negative DPPH and ABTS values documented by all samples might also correspond to the IC_50_ value, because it is inversely proportional to the free radical scavenging activity of the samples, indicating that low IC_50_ samples possessed high antioxidant activity. Overall, the correlation coefficient values of the specimens demonstrated significant (*p* < 0.01) and moderate correlations between TPC-DPPH, TFC-DPPH, TPC-ABTS, TFC-ABTS, and TFC-FRAP.

The DPPH, ABTS, and FRAP R-values of the Pearson correlation coefficient for the TFC were −0.45, −0.39, and 0.42, respectively, revealing that flavonoids were the primary contributor to the antioxidant capacities of DPPH, ABTS, and FRAP. Similar results were observed by Ni et al. [21], with a moderate correlation between TFC and DPPH (R = 0.45) and a strong association between TFC and FRAP (R = 0.81) in *Pleioblastus kongosanensis* f. *aureostriatus* and *Shibataea chinensis*. Moreover, the TPC in this study documented a moderate association with antioxidant activities in DPPH and ABTS, with R-values of −0.40 and −0.42, respectively. Correspondingly, Hu et al. [42] reported a strong relationship between TPC and DPPH (R = 0.74) in *Phyllostachys* spp. leaves. Pande et al. [43] described similar findings for *B. nutans* leaf extracts under different extraction conditions.

The TPC-FRAP revealed no association with the Pearson correlation coefficient, which recorded an R-value of −0.05. The results indicated that TPC did not contribute to the high antioxidant capacity of FRAP. The findings were similar to the report by Ni et al. [21], who demonstrated a weak correlation between TPC and FRAP (R = 0.12). By comparing the correlation coefficient of the R-values, it is possible to suggest that the phenolic and flavonoid groups were slightly responsible for the antioxidant activities of DPPH, ABTS, and FRAP, as stated by Ouyang et al. [44]. In addition, the weaker correlation may be due to the fact that phenolics comprise a sizable collection of chemicals with various structures and antioxidant properties [45]. Due to the presence of non-participating elements such as sugars, flavonoids commonly form bonds with sugar moieties to create glycosides, which have a lower DPPH scavenging activity than their aglycones or phenolic acids on a weight basis [45]. Accordingly, quantifying the contributions of the phenolic and flavonoid compounds to the total antioxidant activity is necessary to understand the correlation between them and their connection to the antioxidant activity.

### 2.4. The BSLA

Statistical analysis of the *B. multiplex* (freeze-drying), *B. tuldoides* (freeze-drying), *B. vulgaris* (microwave-drying), *D. sublaevigata* (freeze-drying), *G. levis* (freeze-drying), and *S. brachycladum* (freeze-drying) samples recorded significant TPC, TFC, DPPH, ABTS, and FRAP values. The bamboo extracts were further studied for their toxicity tests via the BSLA. The LC_50_ (lethal concentration 50) values of the extracts and the positive control, potassium dichromate (K_2_Cr_2_O_7_), are presented in Table 4.

In the current study, the mortality rate of brine shrimp was proportional to the concentration of test samples evaluated. The *B. multiplex*, *B. tuldoides*, *B. vulgaris*, *D. sublaevigata*, *G. levis*, and *S. brachycladum* extracts exhibited no significant toxicity towards brine shrimps at LC_50_ values of 3744.85, 2974.47, 3166.15, 5668.14, 1236.53, and 2045.03 µg/mL, respectively. The positive control, K_2_Cr_2_O_7_, recorded an LC_50_ of 11.23 µg/mL, indicating high toxicity.

The BSLA was conducted to determine the functional properties of the selected bamboo extracts. Nevertheless, reports on the impacts of drying on the toxicity of aqueous bamboo extracts worldwide are limited. Consequently, the present study employed the BSLA as a reliable method for preliminary toxicity assessment of the extracts. Moreover, this study compared the BSLA results of the aqueous bamboo extracts with aqueous medicinal plant leaves extracts.

One investigation observed that among the evaluated shade-dried extracts of *Pentapetes phoenicea*, the chloroform and ethyl acetate extracts were weakly toxic with LC_50_ values of 659.8 and 928.9 μg/mL, respectively [46]. Conversely, the hexane and aqueous extracts were non-toxic, recording LC_50_ values of 1293.6 and 1929.2 μg/mL, respectively [46]. Shawa et al. [47] also reported that an aqueous *Senna singuena* leaves extract air-dried at room temperature did not demonstrate significant toxicity after 24 h. Furthermore, the BSLA values of all *Phragmanthera capitata* leaf solvent extracts (including aqueous extract) were not toxic at LC_50_ > 1000 μg/mL [48].

In conclusion, bamboo extracts could be considered safe for consumption as herbal medicine and could potentially be developed as herbal tea. Nonetheless, the non-toxic attributes exhibited by the plant could be discouraging in treating and managing cancer or tumour alternatives, because BSLA is commonly employed as an indicator for preliminary bioactivity screening, including for anticancer [49].

### 2.5. Chemical Constituents

Traditional medicine uses well-known natural products in the form of secondary metabolites derived from a wide range of natural sources. These specialised metabolites found in fungi, plants, and marine creatures function as a formidable armoury against biotic and abiotic stressors. In addition, medicinal chemists utilise natural products as structural scaffolds to synthesise new medications with enhanced pharmacological efficacy and safety [50,51,52]. Nevertheless, metabolite discovery remains a significant bottleneck in traditional medicine [53]. As a result of multiple erroneous identifications of small compounds, bioactivity investigations of traditional medicines have adopted an evidence-based approach [53]. Thus, LC-MS/MS was used to further quantify *B. multiplex* (freeze-drying), *B. tuldoides* (freeze-drying), *B. vulgaris* (microwave-drying), *D. sublaevigata* (freeze-drying), *G. levis* (freeze-drying), and *S. brachycladum* (freeze-drying) for the purpose of profiling their bioactive compounds that contribute to antioxidant properties and functional pharmaceutical applications.

The liquid chromatogram of the LC-MS/MS analysis for *B. multiplex* is shown in Figure 1. The compounds identified in the *B. multiplex* are tabulated in Table 5 along with their molecular formula, molecular weight, and *m*/*z* value. In the present study, 18 detected peaks showed significance. The highest peak was found to be felodipine (peak 53). In *B. multiplex,* alkaloid compounds identified as caffeine were found in peaks 28, 30, 32, and 37. Other compounds found in *B. multiplex* were: L-histidine (peak 9); pararosaniline (peak 39); felodipine (peak 45); and phytosphingosine (peak 60).

The liquid chromatogram of the LC-MS/MS analysis for *B. tuldoides* is shown in Figure 2. The compounds identified in *B. tuldoides* are tabulated in Table 6 along with their molecular formula, molecular weight, and *m*/*z* value. In the present study, 14 detected peaks showed significance. The highest peak was found to be felodipine (peak 35). In *B. tuldoides*, alkaloid compounds identified as sparteine and papaverine were found in peaks 5 and 16, respectively. Other compounds found in *B. tuldoides* were: PET-cGMP (peak 12); naloxone (peak 14); thiopental (peak 22); cyproheptadine (peak 24); loprazolam (peak 26); difenoconazole (peak 28); RP-8-pCPT-cGMPS (peak 31); and felodipine (peak 44).

The liquid chromatogram of the LC-MS/MS analysis for *B. vulgaris* is shown in Figure 3. The compounds identified in *B. vulgaris* are tabulated in Table 7 along with their molecular formula, molecular weight, and *m*/*z* value. In the present study, 13 detected peaks showed significance. The highest peak was found to be felodipine (peak 42). In *B. vulgaris*, alkaloid compounds identified as papaverine were found in peak 11. Other compounds found in *B. vulgaris* were: econazole (peak 3); pimozide (peak 9); cyproheptadine (peak 15); bisacodyl (peak 19); loprazolam (peak 25); difenoconazole (peak 36); felodipine (peak 51); and cinchocaine (peak 58).

The liquid chromatogram of the LC-MS/MS analysis for *D. sublaevigata* is shown in Figure 4. The compounds identified in *D. sublaevigata* are tabulated in Table 8 along with their molecular formula, molecular weight, and *m*/*z* value. In the present study, 15 detected peaks showed significance. The highest peak was found to be RP-8-pCPT-cGMPS (peak 40). In *D. sublaevigata,* alkaloid compounds identified as papaverine were found in peaks 22 and 24. Other compounds found in *D. sublaevigata* were: phenytoin (peak 5); perazine (peak 12); penconazole (peak 19); cyproheptadine (peak 29); RP-8-pCPT-cGMPS (peak 36); felodipine (peak 49); and cinchocaine (peaks 51 and 56).

The liquid chromatogram of the LC-MS/MS analysis for *G. levis* is shown in Figure 5. The compounds identified in *G. levis* are tabulated in Table 9 along with their molecular formula, molecular weight, and *m*/*z* value. In the present study, 13 detected peaks showed significance. The highest peak was found to be RP-8-pCPT-cGMPS (peak 32). In *G. levis,* alkaloid compounds identified as papaverine were found in peak 15. Other compounds found in *G. levis* were: L-histidine (peak 3); PET-cGMP (peak 10); naloxone (peak 12); cyproheptadine (peak 22); loprazolam (peak 24); RP-8-pCPT-cGMPS (peak 29); felodipine (peak 42); cinchocaine (peak 43); and amphetamine (peak 51).

The liquid chromatogram of the LC-MS/MS analysis for *S. brachycladum* is shown in Figure 6. The compounds identified in *S. brachycladum* are tabulated in Table 10 along with their molecular formula, molecular weight, and *m*/*z* value. In the present study, 13 detected peaks showed significance. The highest peak was found to be felodipine (peak 35). However, no alkaloids were found in *S. brachycladum*. Other compounds found in *S. brachycladum* were: amphetamine (peak 7); naloxone (peak 10); perazine (peak 15); cyproheptadine (peak 24); difenoconazole (peak 27); RP-8-pCPT-cGMPS (peak 32); and felodipine (peak 47).

In brief, on the basis of the notable differences, the alkaloid compounds found in *B. multiplex*, *B. tuldoides*, *B. vulgaris*, *D. sublaevigata*, and *G. levis* were caffeine (Figure 7), papaverine (Figure 8), and sparteine (Figure 9) according to their significant peaks. Through mass spectrometry, caffeine was found in *B. multiplex*, whereas papaverine was found in *B. tuldoides*, *B. vulgaris*, *D. sublaevigata*, and *G. levis*. As for sparteine, it was found only in *B. tuldoides*. Nevertheless, *S. brachycladum* showed no significant peaks attributed to alkaloid compounds.

Previous studies have shown that caffeine and papaverine could influence antioxidant activities. In recent decades, many scientific studies and reviews have documented the interest in caffeine and other coffee bean constituents for their health-promoting properties [54,55]. Many authors have claimed that caffeine is a good antioxidant [56,57,58]. Ren et al. [59] also reported that alkaloids in *Pleioblastus amarus* bamboo shoots were found to contain caffeine when detected through ultra-high-performance liquid chromatography (UHPLC). However, limited research has been conducted on papaverine and its links to antioxidant properties. Interestingly, based on the study by Solmaz et al. [60], in a rat model of sepsis-induced critical illness neuropathy, papaverine exhibited neuroprotective effects due to its anti-inflammatory and antioxidant characteristics. The study suggested that papaverine can contribute to antioxidant properties.

Sparteine is a heterobicyclononane alkaloid with antiarrhythmic properties, which can reduce the incidence of fibrillation and ventricular tachycardia, as well as help regulate blood pressure and heart rate [61]. Additionally, it produces a hypoglycaemic effect and promotes the pancreatic secretion of insulin and glucagon [62]. This alkaloid has also been linked to anti-inflammatory, antimicrobial, diuretic, and uterine contraction-inducing properties [63,64]. Nonetheless, prior research indicated that sparteine lacks antioxidant capabilities.

Non-alkaloids (pharmaceutical compounds) are highlighted in Table 11. Previous studies on biological activities have shown that anticonvulsant drugs contain loprazolam [65], phenytoin [66], and thiopental [67] compounds. Moreover, amphetamine [68], naloxone [69], and perazine [70] have been found in antidepressant drugs. Antifungal drugs also contain difenoconazole [71], econazole [72], and penconazole [73] compounds. Biological activities in antihistamine, antihypertensive, anti-inflammatory, and antipsychotic drugs have been associated with cyproheptadine [74], felodipine [75], L-histidine [76], and pimozide [77] compounds. Stimulant laxatives, antimicrobials, anaesthetic drugs, and dye agents contain bisacodyl [78], phytosphingosine [79], cinchocaine [80], and pararosaniline [81] compounds.

Nevertheless, no phenolic and flavonoid compounds were found in significant peaks, based on the observably low TPC and TFC values of the bamboo extracts. This factor was also observed based on a correlation analysis, as TPC and TFC showed a moderate correlation to antioxidant capacities, thus being slightly responsible for the antioxidant activities. Moreover, using the aqueous extract as a solvent could affect the polarity of compounds, which had a comparable result in alkaloid compounds based on the LC-MS/MS analysis. Yakubu and Bukoye [82] achieved a similar result, because the aqueous extract from *B. vulgaris* leaves was primarily composed of alkaloids, and flavonoids were the least frequent of the phytochemicals. Therefore, it was found that the aqueous bamboo extract profile is high in alkaloid compounds based on LC-MS/MS compared with TPC and TFC. Moreover, the characterisation and optimisation of chemical composition have been highlighted through positive and negative electrospray ionisation (ESI) mode using LC-MS/MS from selected bamboo extracts. According to the results, in contrast to the negative ionisation mode, the positive ionisation mode of LC-MS/MS is appropriate for screening chemical composition in selected bamboo extracts based on the number of significant peaks.

## 3. Materials and Methods

### 3.1. Chemicals and Reagents

The current study utilised acetic acid, anhydrous sodium acetate (NaOAc), anhydrous sodium carbonate (Na_2_CO_3_), hydrochloric acid (HCl), and methanol purchased from Chemiz [United Kingdom (UK)]. Anhydrous aluminium chloride (AlCl_3_), ferric chloride (FeCl_3_) heptahydrate, Folin–Ciocalteu (F-C) reagent, and gallic acid were from Merck (Germany). The study also employed 2,2′-azino-bis(3-ethylbenzothiazoline-6-sulfonic acid) (ABTS) reagent [Sigma-Aldrich, Burlington, MA, United States of America (USA)], dimethyl sulfoxide (DMSO) (Systerm, Selangor, Malaysia), potassium acetate (CH_3_COOK) (R&M, Dundee, UK), potassium dichromate (K_2_Cr_2_O_7_) (Systerm, Malaysia), potassium persulfate (K_2_S_2_O_8_) (HmbG, Hamburg, Germany), quercetin (Targetmol, Boston, MA, USA), Trolox (Targetmol, USA), 2,2-diphenyl-1-picrylhydrazyl (DPPH) reagent (Tokyo Chemical Industry, Tokyo, Japan), and 2,4,6-tris(2-pyridyl)-1,3,5-triazine (TPTZ) reagent (Sigma-Aldrich, USA). The chemicals and reagents were of analytical grades and purchased from Apical Scientific Sdn. Bhd. and Bio3 Scientific Sdn. Bhd., Malaysia, whereas acetonitrile and formic acid derived from Fisher Scientific (USA) were of high-performance liquid chromatography (HPLC) grade and obtained from Syarikat Jaya Usaha, Malaysia.

### 3.2. Plant Materials

The mature leaves of six bamboo species (*Bambusa multiplex*, *B. tuldoides*, *B. vulgaris*, *Dinochloa sublaevigata*, *Gigantochloa levis*, and *Schizostachyum brachycladum*) were harvested between 9.00 a.m. and 11.00 a.m. from the Bamboo Garden, Poring Hot Springs, Ranau, Sabah (6°2′50.795″ N, 116°42′12.731″ E) in May 2021. The samples were collected, placed in zip-lock bags, and identified by an ethnobotanist from the Institute for Tropical Biology and Conservation (ITBC), Universiti Malaysia Sabah (UMS), before being deposited in the BORNEENSIS Gallery, ITBC, UMS. Subsequently, the leaves were cleaned with running tap water and rinsed 4–5 times with distilled water before proceeding to dry treatment.

### 3.3. Drying Process

The samples were harvested at the same time in the morning to ensure metabolite content consistency for the drying techniques comparison. The procedure was based on the methodology suggested by Ni et al. [21] and Chen et al. [83], with some modifications. The drying processes are explained in the following subsections (Table 12). Approximately 50 g of the fresh (control variable) and dried leaves were cut into small pieces and ground into powders with an electric blender (Panasonic, Osaka, Japan). Post-drying, the moisture contents of the samples were below 10%. Subsequently, the samples were packed in different 50 mL centrifuge tubes (Biologix, Camarillo, CA, USA) and stored in a refrigerator at −20 °C (Sharp, Osaka, Japan) before further analyses.

### 3.4. Sample Extraction

The powdered samples were prepared through the lyophilised infusion method suggested by Neményi et al. [84] with minor modifications. First, a portion of each dried sample (1 g) was infused with 50 mL distilled water (100 °C) for 5 min at room temperature (22 °C). Subsequently, the mixtures were filtered with filter papers (Whatman, Maidstone, UK) and preserved in a 50 mL centrifuge tube (Biologix, USA). The resulting infusions were then lyophilised in a freeze-dryer (Labconco, USA) for 2.5 days to remove excess water, as Valentão et al. [85] proposed, with minor modifications. The lyophilised aqueous extract yields were stored in a refrigerator at 4 °C (Sharp, Japan). Before assessments, the extracts were adjusted to the appropriate concentration in distilled water.

### 3.5. Phytochemical Analysis

#### 3.5.1. Determination of TPC

The TPC of the samples in this study was determined with the Folin–Ciocalteu method as outlined by Ainsworth and Gillespie [86], with some modifications. Aqueous gallic acid solutions (100 µg/mL) were employed as standards for the calibration curve (Appendix A). In each replicate, 100 µL of relevantly diluted standard solutions, 200 µL of 10% (*v*/*v*) F–C reagent, and 800 µL of 700 mM anhydrous Na_2_CO_3_ were mixed and vortexed. Subsequently, the mixtures were incubated for 2 h in the dark at room temperature. The absorbance of the standard was measured in a 96-well culture plate (Biologix, USA) at 765 nm against a blank (distilled water) with a microplate reader (Multiskan SkyHigh, Thermo Fisher Scientific, Waltham, MA, USA). Similarly, 100 µL of the sample extracts were reacted with F–C reagent and Na_2_CO_3_ to determine their phenolic contents. The results were expressed as the mg of gallic acid equivalent to 1 g of the dried sample (mg GAE/g).

#### 3.5.2. Determination of TFC

In the present study, the TFCs of the bamboo samples were determined with the aluminium chloride colorimetric method as reported by Chang et al. [87], with minor alterations. Quercetin was utilised in obtaining the calibration curve (Appendix A), where 10 mg of the substance was dissolved in distilled water and diluted to 100 µg/mL. In each replicate, 120 µL of the diluted standard solution was separately mixed with 360 µL methanol (95%), 24 µL anhydrous AlCl_3_ [10% (*w*/*v*)], 24 µL CH_3_COOK (1 M), and 680 µL distilled water. After incubation at room temperature for 30 min, the reaction mixtures were added to a 96-well culture plate (Biologix, USA) before measuring the absorbance at 415 nm against the blank (distilled water) with a microplate reader (Multiskan SkyHigh, Thermo Fisher Scientific, USA). A similar procedure was followed with 120 µL of the sample extracts reacting with methanol, AlCl_3_, CH_3_COOK, and distilled water to determine their flavonoid contents. The results were expressed as mg of quercetin equivalent to 1 g of the dried sample (mg QE/g).

### 3.6. Antioxidant Analysis

#### 3.6.1. Determination of DPPH

The DPPH free radical scavenging activities of the extracts were determined according to the report by Chan et al. [88], with slight alterations. First, at respective concentrations, 50 µL of the plant extracts were reacted with 195 µL DPPH–methanolic solution (0.1 mM) in a 96-well culture plate (Biologix, USA). The mixtures were then swirled gently for 1 min and allowed to stand for 1 h. Finally, the absorbance of the resulting mixture was measured with a microplate reader (Multiskan SkyHigh, Thermo Fisher Scientific, USA) at 540 nm against the blank (distilled water). Trolox was employed as the antioxidant reference standard within the 6.25–100 µg/mL concentration range (Appendix A). The findings were expressed as IC_50_ values (the sample concentration required to inhibit 50% of DPPH radicals) by extrapolating the regression analysis.

#### 3.6.2. Determination of ABTS

The antioxidant capacities of the samples in this study were measured according to the ABTS free radical scavenging activity procedure adopted by Lee et al. [89], with minor modifications. First, the ABTS was prepared by reacting 5 mL of 7 mM ABTS water solution with 88 µL of 140 mM K_2_S_2_O_8_ at a 1:0.35 ratio. Subsequently, the mixture was allowed to stand in the dark at room temperature for 16 h. Before performing the assay, the ABTS stock solution was diluted with distilled water (at a 1:88 ratio) to obtain an absorbance at 734 nm (0.70 ± 0.02) and equilibrated to 30 °C.

The scavenging activities of the bamboo leaves in the current study were determined by mixing 100 µL of the samples with 100 µL of ABTS reagent in a 96-well culture plate (Biologix, USA) and incubating at room temperature for 6 min. After incubation, the absorbance was measured at 734 nm against the blank (distilled water) with a microplate reader (Multiskan SkyHigh, Thermo Fisher Scientific, USA). Trolox was employed as the antioxidant reference standard within the 6.25–100 µg/mL range (Appendix A). The IC_50_ ABTS values (the sample concentration required to inhibit 50% of the ABTS radicals) were procured by extrapolating the regression analysis results.

#### 3.6.3. Determination of FRAP

The FRAP assay performed in the present study was based on the slightly modified procedure described by Russo et al. [90]. The FRAP reagent was prepared by mixing 38 mM anhydrous NaOAc in distilled water, pH 3.6, with 20 mM FeCl_3_ heptahydrate in distilled water and 10 mM TPTZ in 40 mM HCl in a 10:1:1 ratio. Approximately 20 µL of the leaf extracts and 180 µL of FRAP reagent were mixed in a 96-well culture plate (Biologix, USA) and incubated at 37 °C in a water bath (Daihan Scientific, Wonju, South Korea) for 40 min in the dark. As blanks, 20 µL distilled water was added to the 180 µL FRAP reagent. The absorbances of the resultant mixtures were measured at 593 nm against the blank (distilled water) with a microplate reader (Multiskan SkyHigh, Thermo Fisher Scientific, USA). Trolox was utilised as the antioxidant reference standard within the 0–100 µg/mL concentration range (Appendix A). The values were communicated as mg of Trolox equivalent to 1 g of dried sample (mg TE/g).

### 3.7. Determination of BSLA

The BSLA was determined according to the guidelines reported by Rajeh et al. [91], with some modifications. The toxicity of the compounds was assessed at 1000, 100, 10, and 1 µg/mL in 10 mL seawater solutions with 1% DMSO (*v*/*v*). In the current study, the brine shrimp cysts were hatched in a small aquarium containing natural seawater (pH 8.0) for approximately 48 h under aeration with continuous illumination at 25 °C. The nauplii were lured to one side of the vessel with a light source to isolate them. Subsequently, active nauplii were collected for examination with a plastic pipette after hatching. After 48 h of development, 10 nauplii were transplanted to each plate, and the number of survivors was counted after 24 h.

After the addition of the samples, the plates were incubated at 25 °C for 24 h. The specimens were then dissolved in DMSO at a maximum concentration of 2% to prevent potential toxicity from the solvent [92,93]. K_2_Cr_2_O_7_ served as the positive control. After 24 h, the plates were examined under a binocular microscope (12.5× magnification) to determine the number of survivors, thus obtaining the mortality percentage. Subsequently, the nauplii were killed with methanol, and the number of dead (immobile nauplii) in each well was recorded. The chronic LC_50_, or the lethal concentration resulting in 50% death after 24 h of exposure, was measured with the probit method to measure the toxicity of the extracts. The LC_50_ data were then determined from the regression line produced by extrapolating the concentration with the percentage of fatalities on a probit scale. Finally, each outcome was tabulated and analysed.

### 3.8. LC-MS/MS Analysis

#### 3.8.1. Sample Preparation

The bamboo extracts were dissolved with ultra-purified water (total oxidisable carbon ≤5 ppb and resistivity of 18.2 MΩ-cm) in a 50 mL centrifuge tube (Biologix, USA). The extracts were then sonicated for 15 min (20–22 °C), filtered using syringe filters (polytetrafluoroethylene filter, pore size of 0.04 µm, Merck, Darmstadt, Germany), and transferred into HPLC vials (ChromineX, Selangor, Malaysia) for further analysis.

#### 3.8.2. LC-ESI-QTOF-MS/MS Parameters

The protocol developed by Gu et al. [94] was adopted for the chemical profiling of bamboo extracts, with slight modifications. Chemical compounds in the extracts were detected by LC-QTOF-MS/MS (liquid chromatography–quadrupole time-of-flight tandem mass spectrometry) (Bruker impact II, Bruker Daltonics, Bremen, Germany). The positive ionisation mode was applied to obtain high-resolution spectra of compounds. On the Thermo UltiMate 3000 HPLC system (Thermo Fisher Scientific, USA), the C_18_ column (3 × 150 mm, 3 µm particle size, Acclaim Polar Advantage II, Thermo Fisher Scientific, USA) was used for the LC separation. ESI was performed with the following settings: capillary voltage of 4500 V; drying gas of 10.0 L/min at 250 °C; endplate offset of −500 V; mass range of 50–1500 *m*/*z*; and nebulizer pressure of 2.0 bar. High-resolution MS was carried out using the Bruker impact II QTOF instrument (Bruker Daltonics, Germany). The TOF system was programmed using the following settings: corrector fill of 71.4 V; reflector of 2600.0 V; flight tube of 9900.0 V; corrector extract of 400.0 V; and detector of 2226.6 V.

Mobile phase A consisted of water/formic acid (99:1, *v*/*v*), whereas mobile phase B was composed of acetonitrile/formic acid (99:1, *v*/*v*). Formic acid was used because of its compatibility with MS analysis. Both the A and B mobile phases were degassed at 21 °C for 15 min. The injection volume for each sample was 6 µL, and the flow rate was adjusted to 0.8 mL/min. Using a mixture of both the A and B mobile phases, gradient elution was carried out as follows: 0–20 min, 10% B; 20–30 min, 25% B; 30–40 min, 35% B; 40–70 min, 40% B; 70–75 min, 55% B; 75–77 min, 80% B; 77–79 min, 100% B; 79–82 min, 100% B; and 82–85 min, 10% B. At the conclusion of the programme, the eluent content was restored to the baseline gradient and the column was equilibrated for 3 min prior to the subsequent injection.

#### 3.8.3. Data Processing

The MS raw data were obtained using Bruker Compass DataAnalysis version 4.2 (Bruker Daltonics, Germany). By applying the advanced libraries (ESI MSn Lib, Pharmaceuticals, and Plant Metabolites databases) in the system, the chemical compounds were generated by detecting and comparing the mass spectra of the samples with molecular weights based on the library databases adjusted for positive ionisation with detailed mass spectra confirmation (Appendix A). The significant peaks were then tabulated and examined.

### 3.9. Statistical Analysis

Each procedure in the current study was conducted in triplicates and the data were expressed as mean ± standard deviation (SD). Statistical comparisons were performed with one-way analysis of variance (ANOVA), followed by Duncan’s post hoc test, with the level of statistical significance set at *p* < 0.05. Correlation analysis was performed via the Pearson correlation coefficient (r). The Statistical Package for Social Sciences (SPSS) for Windows (Version 19.0, IBM Corporation, New York, NY, USA) was employed in the statistical analyses.

## 4. Conclusions

The results of the different drying techniques documented significant differences (*p* < 0.05), indicating that microwave-, oven-, and freeze-drying retained superior TPC, TFC, DPPH, ABTS, and FRAP to the conventional methods of sun- and shade-drying. Furthermore, the correlation between the TFC and TPC phytochemical contents and the DPPH, ABTS, and FRAP antioxidant capacities exhibited a moderate association, suggesting that the TPC and TFC had a minor contribution to the antioxidant activity. Freeze-drying was recorded significantly better in all bamboo species, excluding *B. vulgaris*, which favoured microwave-drying. Although an expensive and energy-intensive technology, freeze-drying produced better-quality products in terms of preserving the antioxidant potential. Moreover, the LC_50_ results at >1000 µg/mL obtained in the BSLA demonstrated no toxicity, indicating that the bamboo extracts were safe to be consumed. The LC-MS results show that alkaloid and pharmaceutical compounds have been found in the extracts, based on the significant peaks in the chromatograms. Thus, in line with past studies, these compounds may stimulate antioxidant properties. This discovery may assist in further research into developing bamboo leaves as functional food items, such as bamboo tea. The investigation of bamboo extracts for medicinal components may also contribute to the search for potential drugs.

## Figures and Tables

**Figure 1 molecules-27-06458-f001:**
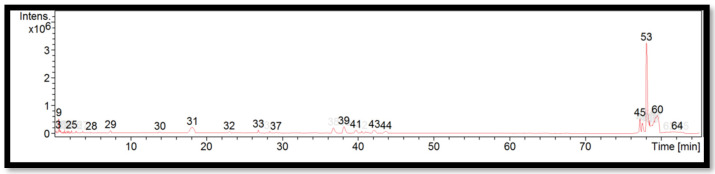
LC-MS/MS chromatogram of *B. multiplex*.

**Figure 2 molecules-27-06458-f002:**
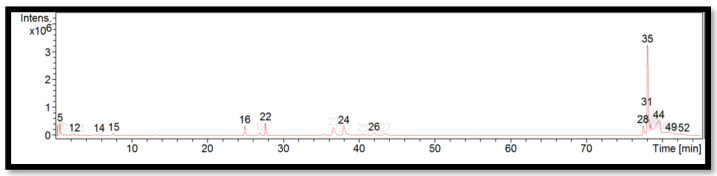
LC-MS/MS chromatogram of *B. tuldoides*.

**Figure 3 molecules-27-06458-f003:**
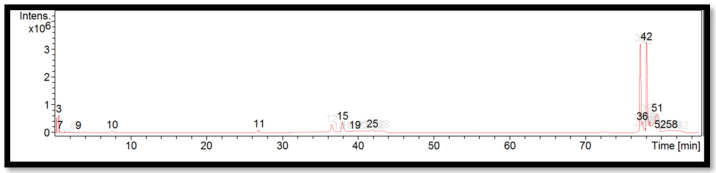
LC-MS/MS chromatogram of *B. vulgaris*.

**Figure 4 molecules-27-06458-f004:**
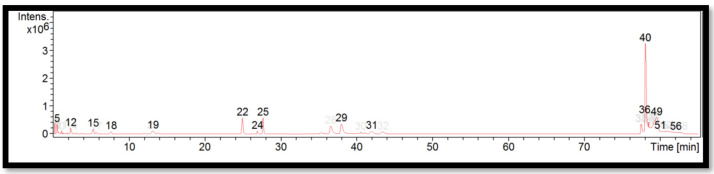
LC-MS/MS chromatogram of *D. sublaevigata*.

**Figure 5 molecules-27-06458-f005:**
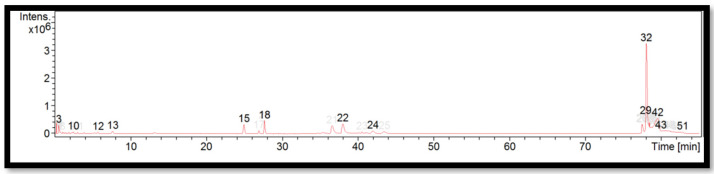
LC-MS/MS chromatogram of *G. levis*.

**Figure 6 molecules-27-06458-f006:**
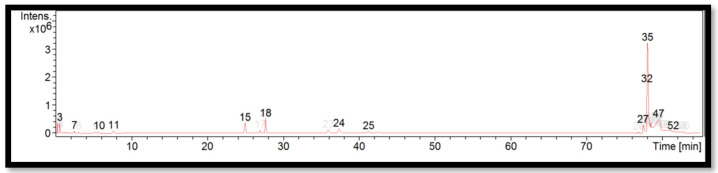
LC-MS/MS chromatogram of *S. brachycladum*.

**Figure 7 molecules-27-06458-f007:**
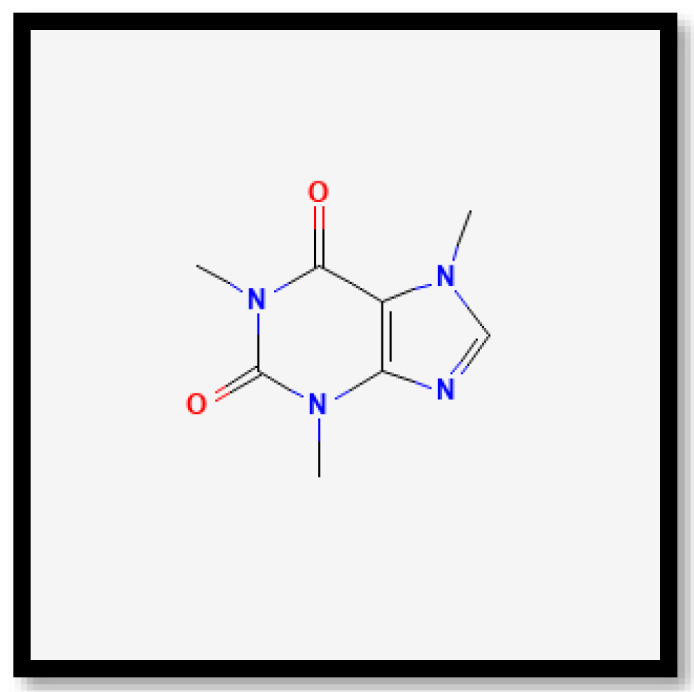
Chemical structures of caffeine.

**Figure 8 molecules-27-06458-f008:**
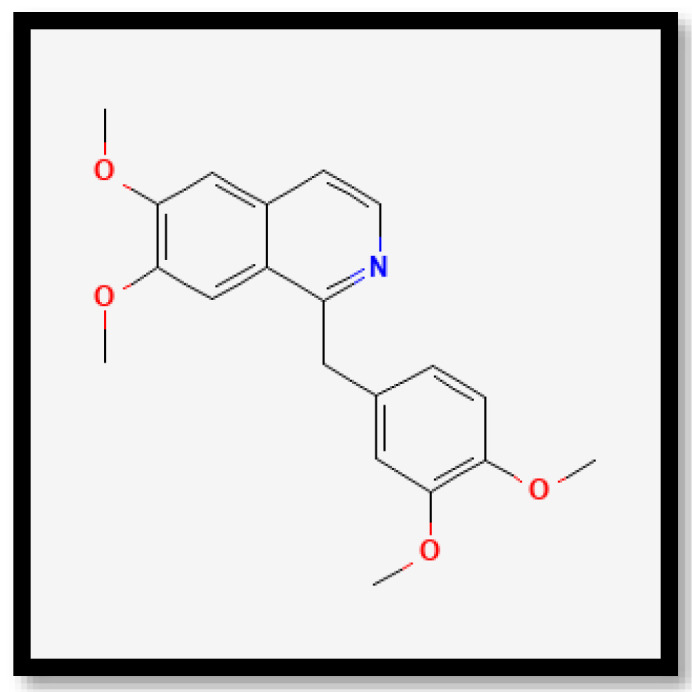
Chemical structures of papaverine.

**Figure 9 molecules-27-06458-f009:**
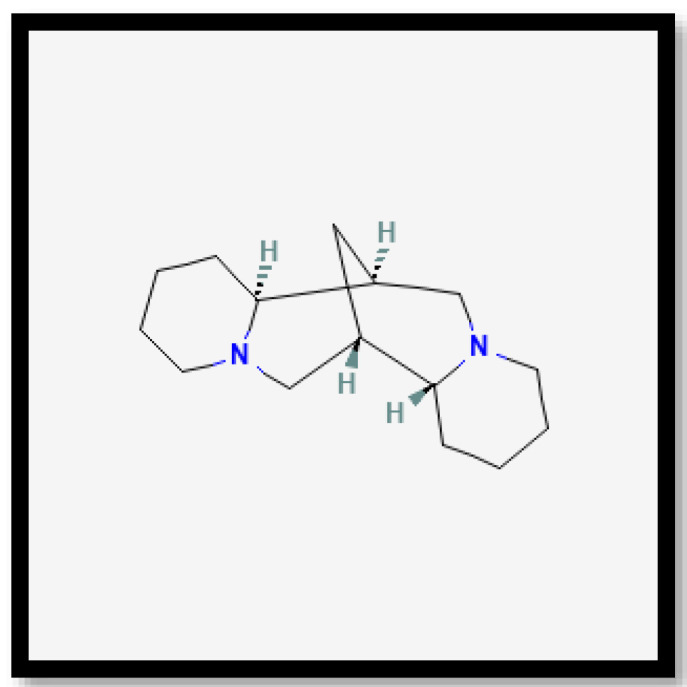
Chemical structures of sparteine.

**Table 1 molecules-27-06458-t001:** The TPC and TFC of the selected bamboo extracts dried with different methods.

Drying Methods	TPC ^1^	TFC ^2^
*B. multiplex*		
Fresh ^3^	5.64 ± 0.09 ^b^	0.87 ± 0.01 ^f^
Sun-drying	5.09 ± 0.01 ^d^	1.13 ± 0.00 ^c^
Shade-drying	5.42 ± 0.02 ^c^	1.26 ± 0.00 ^b^
Microwave-drying	5.44 ± 0.05 ^c^	1.04 ± 0.01 ^d^
Oven-drying	5.18 ± 0.00 ^d^	0.89 ± 0.00 ^e^
Freeze-drying	5.74 ± 0.06 ^a^	1.62 ± 0.01 ^a^
*B. tuldoides*		
Fresh ^3^	5.26 ± 0.01 ^d^	1.65 ± 0.01 ^d^
Sun-drying	4.31 ± 0.03 ^f^	1.17 ± 0.00 ^f^
Shade-drying	4.92 ± 0.01 ^e^	1.41 ± 0.00 ^e^
Microwave-drying	5.91 ± 0.00 ^a^	2.06 ± 0.00 ^b^
Oven-drying	5.60 ± 0.04 ^c^	1.69 ± 0.02 ^c^
Freeze-drying	5.84 ± 0.01 ^b^	2.11 ± 0.00 ^a^
*B. vulgaris*		
Fresh ^3^	4.64 ± 0.05 ^e^	0.84 ± 0.00 ^d^
Sun-drying	4.24 ± 0.00 ^f^	0.81 ± 0.00 ^e^
Shade-drying	5.45 ± 0.00 ^d^	0.85 ± 0.00 ^d^
Microwave-drying	6.17 ± 0.04 ^a^	1.10 ± 0.01 ^a^
Oven-drying	5.77 ± 0.03 ^b^	0.94 ± 0.01 ^c^
Freeze-drying	5.58 ± 0.01 ^c^	0.96 ± 0.00 ^b^
*D. sublaevigata*		
Fresh ^3^	8.26 ± 0.05 ^b^	0.78 ± 0.00 ^b,c^
Sun-drying	7.38 ± 0.00 ^d^	0.78 ± 0.00 ^b^
Shade-drying	8.27 ± 0.09 ^b^	0.77 ± 0.00 ^c^
Microwave-drying	8.20 ± 0.02 ^b^	0.83 ± 0.00 ^a^
Oven-drying	7.91 ± 0.01 ^c^	0.84 ± 0.00 ^a^
Freeze-drying	12.59 ± 0.09 ^a^	0.84 ± 0.00 ^a^
*G. levis*		
Fresh ^3^	3.92 ± 0.00 ^d^	0.81 ± 0.01 ^c^
Sun-drying	2.69 ± 0.01 ^f^	0.77 ± 0.01 ^d^
Shade-drying	3.68 ± 0.02 ^e^	0.84 ± 0.00 ^b^
Microwave-drying	4.60 ± 0.03 ^b^	0.82 ± 0.01 ^c^
Oven-drying	4.29 ± 0.00 ^c^	0.84 ± 0.00 ^b^
Freeze-drying	4.78 ± 0.01 ^a^	0.87 ± 0.01 ^a^
*S. brachycladum*		
Fresh ^3^	4.34 ± 0.09 ^d^	1.85 ± 0.01 ^b^
Sun-drying	4.23 ± 0.01 ^e^	0.98 ± 0.00 ^f^
Shade-drying	4.34 ± 0.04 ^d^	1.31 ± 0.01 ^e^
Microwave-drying	5.01 ± 0.03 ^c^	1.70 ± 0.00 ^d^
Oven-drying	5.30 ± 0.09 ^b^	1.74 ± 0.00 ^c^
Freeze-drying	5.61 ± 0.01 ^a^	2.12 ± 0.01 ^a^

The values represent the means ± standard deviations of three replicates. Different letters (within a column) indicate significant differences (one-way ANOVA, Duncan’s multiple comparison test, *p* < 0.05). ^1^ TPC was expressed as mg gallic acid equivalent to 1 g of dried sample (mg GAE/g). ^2^ TFC was expressed as the mg quercetin equivalent to 1 g of dried sample (mg QE/g). ^3^ Fresh sample was expressed as control variable.

**Table 2 molecules-27-06458-t002:** The DPPH, ABTS, and FRAP assay results of the selected bamboo extracts dried with different methods.

Drying Methods	DPPH ^1^	ABTS ^2^	FRAP ^3^
*B. multiplex*			
Fresh ^4^	3.73 ± 0.00 ^e^	2.93 ± 0.01 ^e^	31.33 ± 0.05 ^d^
Sun-drying	4.09 ± 0.00 ^f^	2.77 ± 0.01 ^d^	22.39 ± 0.11 ^f^
Shade-drying	3.42 ± 0.00 ^b^	2.59 ± 0.01 ^b^	29.73 ± 0.07 ^e^
Microwave-drying	3.67 ± 0.01 ^d^	2.72 ± 0.01 ^c^	33.83 ± 0.10 ^b^
Oven-drying	3.49 ± 0.00 ^c^	2.50 ± 0.01 ^a^	32.06 ± 0.07 ^c^
Freeze-drying	3.20 ± 0.00 ^a^	2.73 ± 0.00 ^c^	36.65 ± 0.09 ^a^
*B. tuldoides*			
Fresh ^4^	3.54 ± 0.02 ^d^	3.07 ± 0.01 ^f^	27.68 ± 0.12 ^d^
Sun-drying	4.01 ± 0.00 ^f^	2.70 ± 0.00 ^e^	19.40 ± 0.11 ^f^
Shade-drying	3.82 ± 0.01 ^e^	2.60 ± 0.01 ^d^	25.87 ± 0.10 ^e^
Microwave-drying	3.37 ± 0.01 ^c^	2.29 ± 0.01 ^c^	35.98 ± 0.06 ^a^
Oven-drying	3.00 ± 0.02 ^b^	1.89 ± 0.01 ^a^	33.83 ± 0.01 ^c^
Freeze-drying	2.92 ± 0.01 ^a^	1.98 ± 0.00 ^b^	35.83 ± 0.05 ^b^
*B. vulgaris*			
Fresh ^4^	3.59 ± 0.01 ^c^	2.74 ± 0.00 ^d^	28.36 ± 0.14 ^d^
Sun-drying	4.14 ± 0.00 ^f^	2.94 ± 0.00 ^f^	15.86 ± 0.10 ^f^
Shade-drying	3.72 ± 0.00 ^e^	2.80 ± 0.01 ^e^	25.16 ± 0.13 ^e^
Microwave-drying	3.11 ± 0.00 ^a^	2.32 ± 0.00 ^c^	35.32 ± 0.15 ^a^
Oven-drying	3.37 ± 0.00 ^b^	2.01 ± 0.00 ^a^	33.47 ± 0.10 ^c^
Freeze-drying	3.63 ± 0.00 ^d^	2.13 ± 0.01 ^b^	34.74 ± 0.14 ^b^
*D. sublaevigata*			
Fresh ^4^	3.78 ± 0.00 ^d^	2.57 ± 0.00 ^c^	14.06 ± 0.10 ^e^
Sun-drying	4.33 ± 0.00 ^f^	2.76 ± 0.00 ^d^	6.40 ± 0.12 ^f^
Shade-drying	4.12 ± 0.00 ^e^	2.58 ± 0.01 ^c^	15.50 ± 0.01 ^d^
Microwave-drying	3.36 ± 0.00 ^c^	2.38 ± 0.01 ^b^	19.50 ± 0.12 ^c^
Oven-drying	3.16 ± 0.00 ^b^	2.22 ± 0.00 ^a^	24.60 ± 0.06 ^b^
Freeze-drying	3.05 ± 0.00 ^a^	2.38 ± 0.01 ^b^	31.23 ± 0.11 ^a^
*G. levis*			
Fresh ^4^	4.42 ± 0.01 ^f^	3.09 ± 0.00 ^d^	29.35 ± 0.14 ^c^
Sun-drying	4.36 ± 0.00 ^e^	3.47 ± 0.00 ^f^	18.26 ± 0.17 ^e^
Shade-drying	4.11 ± 0.01 ^d^	3.22 ± 0.01 ^e^	22.80 ± 0.09 ^d^
Microwave-drying	4.05 ± 0.00 ^c^	3.05 ± 0.01 ^c^	34.22 ± 0.11 ^b^
Oven-drying	3.37 ± 0.00 ^b^	2.47 ± 0.00 ^b^	34.85 ± 0.09 ^a^
Freeze-drying	3.23 ± 0.00 ^a^	2.44 ± 0.00 ^a^	35.02 ± 0.07 ^a^
*S. brachycladum*			
Fresh ^4^	3.49 ± 0.01 ^c^	2.41 ± 0.00 ^c^	27.40 ± 0.07 ^d^
Sun-drying	4.73 ± 0.02 ^f^	3.10 ± 0.01 ^e^	13.33 ± 0.03 ^e^
Shade-drying	4.45 ± 0.01 ^e^	3.33 ± 0.01 ^f^	13.18 ± 0.02 ^f^
Microwave-drying	3.71 ± 0.01 ^d^	2.36 ± 0.00 ^b^	30.22 ± 0.08 ^b^
Oven-drying	3.28 ± 0.01 ^b^	2.12 ± 0.01 ^a^	28.03 ± 0.06 ^c^
Freeze-drying	3.23 ± 0.00 ^a^	2.51 ± 0.00 ^d^	35.81 ± 0.09 ^a^
Trolox ^5^	4.09 ± 0.00	4.55 ± 0.02	–

The values represent the means ± standard deviations of three replicates. Different letters (within a column) indicate significant differences (one-way ANOVA, Duncan’s multiple comparison test, *p* < 0.05). ^1^ DPPH is expressed as IC_50_ (µg/mL). ^2^ ABTS is expressed as IC_50_ (µg/mL). ^3^ FRAP is expressed as mg Trolox equivalent to 1 g of dried sample (mg TE/g). ^4^ Fresh sample is expressed as a control variable. ^5^ Trolox is expressed as a positive control.

**Table 3 molecules-27-06458-t003:** The Pearson correlation between the phytochemical contents and antioxidant capacities of the dried bamboo extracts.

Phytochemical	Antioxidant Capacity					
	DPPH		ABTS		FRAP	
	R	*p*-Value	R	*p*-Value	R	*p*-Value
TPC	−0.40 **	0.00	−0.42 **	0.00	−0.05	0.64
TFC	−0.45 **	0.00	−0.39 **	0.00	0.42 **	0.00

** Correlation is significant at the 0.01 level (2-tailed).

**Table 4 molecules-27-06458-t004:** The mortality percentage and lethality concentration of shrimp nauplii after treatment with the bamboo extracts.

Samples	Concentration (µg/mL)	% Mortality	LC_50_ (µg/mL)
K_2_Cr_2_O_7_ ^1^	1000	100	11.23
	100	33	
	10	33	
	1	27	
*B. multiplex*	1000	17	3744.85
	100	10	
	10	7	
	1	0	
*B. tuldoides*	1000	20	2974.47
	100	10	
	10	7	
	1	0	
*B. vulgaris*	1000	17	3166.15
	100	13	
	10	7	
	1	0	
*D. sublaevigata*	1000	13	5668.14
	100	10	
	10	7	
	1	0	
*G. levis*	1000	27	1236.53
	100	20	
	10	10	
	1	0	
*S. brachycladum*	1000	20	2045.03
	100	17	
	10	10	
	1	0	

^1^ K_2_Cr_2_O_7_ was expressed as a positive control.

**Table 5 molecules-27-06458-t005:** The compounds identified in *B. multiplex*.

Peak	RT (min)	Identified Compounds	Molecular Formula	Molecular Weight	*m*/*z*
3	0.5	Unidentified	–	199.9663	200.9736
9	0.6	L-Histidine	C_6_H_9_N_3_O_2_	155.0354	156.0427
25	2.2	Unidentified	–	445.2898	446.2971
28	4.8	Caffeine	C_8_H_10_N_4_O_2_	577.3688	578.3760
29	7.4	Unidentified	–	452.3372	453.3445
30	13.9	Caffeine	C_8_H_10_N_4_O_2_	550.1329	551.1402
31	18.2	Unidentified	–	534.1379	535.1452
32	23.0	Caffeine	C_8_H_10_N_4_O_2_	534.1378	535.1451
33	26.9	Unidentified	–	700.4868	701.4941
37	29.2	Caffeine	C_8_H_10_N_4_O_2_	428.1840	429.1913
39	38.2	Pararosaniline	C_19_H_17_N_3_	287.2837	288.2909
41	39.7	Unidentified	–	315.2784	316.2857
43	42.2	Unidentified	–	315.3145	316.3218
44	43.7	Unidentified	–	315.3144	316.3217
45	77.2	Felodipine	C_18_H_19_Cl_2_NO_4_	337.3346	338.3419
53	78.0	Felodipine	C_18_H_19_Cl_2_NO_4_	337.3365	338.3438
60	79.5	Phytosphingosine	C_18_H_39_NO_3_	337.3346	338.3419
64	82.1	Unidentified	–	343.2721	344.2794

**Table 6 molecules-27-06458-t006:** The compounds identified in *B. tuldoides*.

Peak	RT (min)	Identified Compounds	Molecular Formula	Molecular Weight	*m*/*z*
5	0.6	Sparteine	C_15_H_26_N_2_	234.1583	235.1656
12	2.5	PET-cGMP	C_18_H_15_N_5_O_7_PNa	113.0840	114.0913
14	5.7	Naloxone	C_19_H_21_NO_4_	327.2521	328.2594
15	7.6	Unidentified	–	452.3361	453.3434
16	25.0	Papaverine	C_20_H_21_NO_4_	678.5037	340.2591
22	27.7	Thiopental	C_11_H_18_N_2_O_2_S	791.5873	396.8009
24	38.0	Cyproheptadine	C_21_H_21_N	287.2827	288.2900
26	42.0	Loprazolam	C_23_H_21_ClN_6_O_3_	315.3139	316.3211
28	77.5	Difenoconazole	C_19_H_17_Cl_2_N_3_O_3_	309.3035	310.3108
31	78.0	RP-8-pCPT-cGMPS	C_16_H_14_ClN_5_O_6_PS_2_Na	311.3193	312.3265
35	78.0	Felodipine	C_18_H_19_Cl_2_NO_4_	337.3367	338.3439
44	79.5	Felodipine	C_18_H_19_Cl_2_NO_4_	337.3348	338.3421
49	81.2	Unidentified	–	337.3347	338.3419
52	82.8	Unidentified	–	225.9441	226.9514

**Table 7 molecules-27-06458-t007:** The compounds identified in *B. vulgaris*.

Peak	RT (min)	Identified Compounds	Molecular Formula	Molecular Weight	*m*/*z*
3	0.5	Econazole	C_18_H_15_Cl_3_N_2_O	155.0349	156.0422
7	0.7	Unidentified	–	200.0322	201.0394
9	3.1	Pimozide	C_28_H_29_F_2_N_3_O	489.3156	490.3229
10	7.6	Unidentified	–	452.3377	453.3449
11	26.9	Papaverine	C_20_H_21_NO_4_	678.5046	340.2596
15	37.9	Cyproheptadine	C_21_H_21_N	287.2832	288.2905
19	39.6	Bisacodyl	C_22_H_19_NO_4_	361.1714	362.1787
25	41.9	Loprazolam	C_23_H_21_ClN_6_O_3_	315.3143	316.3216
36	77.5	Difenoconazole	C_19_H_17_Cl_2_N_3_O_3_	337.3351	338.3423
42	78.0	Felodipine	C_18_H_19_Cl_2_NO_4_	320.3085	321.3158
51	79.5	Felodipine	C_18_H_19_Cl_2_NO_4_	337.3352	338.3425
52	79.8	Unidentified	–	343.2730	344.2802
58	81.4	Cinchocaine	C_20_H_29_N_3_O_2_	337.3349	338.3422

**Table 8 molecules-27-06458-t008:** The compounds identified in *D. sublaevigata*.

Peak	RT (min)	Identified Compounds	Molecular Formula	Molecular Weight	*m*/*z*
5	0.5	Phenytoin	C_15_H_12_N_2_O_2_	155.0349	156.0421
12	2.4	Perazine	C_20_H_25_N_3_S	339.2523	340.2596
15	5.3	Unidentified	–	474.3177	475.3250
18	7.7	Unidentified	–	452.3365	453.3438
19	13.2	Penconazole	C_13_H_15_Cl_2_N_3_	565.4204	566.4276
22	25.0	Papaverine	C_20_H_21_NO_4_	678.5038	679.5111
24	27.0	Papaverine	C_20_H_21_NO_4_	678.5026	340.2586
25	27.7	Unidentified	–	813.5703	814.5776
29	38.0	Cyproheptadine	C_21_H_21_N	287.2825	288.2897
31	42.0	Unidentified	–	315.3136	316.3209
36	78.0	RP-8-pCPT-cGMPS	C_16_H_14_ClN_5_O_6_PS_2_Na	311.3185	312.3258
40	78.0	RP-8-pCPT-cGMPS	C_16_H_14_ClN_5_O_6_PS_2_Na	320.3077	321.3149
49	79.5	Felodipine	C_18_H_19_Cl_2_NO_4_	337.3339	338.3412
51	80.0	Cinchocaine	C_20_H_29_N_3_O_2_	343.2714	344.2787
56	82.1	Cinchocaine	C_20_H_29_N_3_O_2_	343.2717	344.2789

**Table 9 molecules-27-06458-t009:** The compounds identified in *G. levis*.

Peak	RT (min)	Identified Compounds	Molecular Formula	Molecular Weight	*m*/*z*
3	0.5	L-Histidine	C_6_H_9_N_3_O_2_	155.0347	156.0420
10	2.5	PET-cGMP	C_18_H_15_N_5_O_7_PNa	113.0840	114.0913
12	5.7	Naloxone	C_19_H_21_NO_4_	327.2530	328.2602
13	7.6	Unidentified	–	452.3369	453.3442
15	25.0	Papaverine	C_20_H_21_NO_4_	678.5056	340.2601
18	27.7	Unidentified	–	813.5731	814.5803
22	38.0	Cyproheptadine	C_21_H_21_N	287.2833	288.2906
24	42.0	Loprazolam	C_23_H_21_ClN_6_O_3_	315.3148	316.3221
29	78.0	RP-8-pCPT-cGMPS	C_16_H_14_ClN_5_O_6_PS_2_Na	311.3195	312.3268
32	78.0	RP-8-pCPT-cGMPS	C_16_H_14_ClN_5_O_6_PS_2_Na	674.6705	675.6778
42	79.5	Felodipine	C_18_H_19_Cl_2_NO_4_	337.3351	338.3424
43	80.0	Cinchocaine	C_20_H_29_N_3_O_2_	343.2728	344.2801
51	82.8	Amphetamine	C_9_H_13_N	225.9441	226.9513

**Table 10 molecules-27-06458-t010:** The compounds identified in *S. brachycladum*.

Peak	RT (min)	Identified Compounds	Molecular Formula	Molecular Weight	*m*/*z*
3	0.5	Unidentified	–	155.0348	156.0421
7	2.5	Amphetamine	C_9_H_13_N	113.0840	114.0913
10	5.7	Naloxone	C_19_H_21_NO_4_	327.2523	328.2596
11	7.6	Unidentified	–	452.3367	453.3440
15	25.0	Perazine	C_20_H_25_N_3_S	678.5043	340.2594
18	27.7	Unidentified	–	829.5351	415.7748
24	37.4	Cyproheptadine	C_21_H_21_N	287.2828	288.2901
25	41.3	Unidentified	–	315.3141	316.3214
27	77.5	Difenoconazole	C_19_H_17_Cl_2_N_3_O_3_	309.3037	310.3110
32	78.0	RP-8-pCPT-cGMPS	C_16_H_14_ClN_5_O_6_PS_2_Na	311.3194	312.3267
35	78.0	Felodipine	C_18_H_19_Cl_2_NO_4_	337.3365	338.3437
47	79.5	Felodipine	C_18_H_19_Cl_2_NO_4_	337.3351	338.3423
52	81.4	Unidentified	–	343.2726	344.2799

**Table 11 molecules-27-06458-t011:** Non-alkaloid compounds with their properties.

Chemical Compounds	Properties	References
Loprazolam	Anticonvulsant drugs	McDonough Jr et al. [65]
Phenytoin	Anticonvulsant drugs	Mishory et al. [66]
Thiopental	Anticonvulsant drugs	Papatheodoropoulos et al. [67]
Amphetamine	Antidepressant drugs	Stahl [68]
Naloxone	Antidepressant drugs	Sikka et al. [69]
Perazine	Antidepressant drugs	Wójcikowski and Daniel [70]
Difenoconazole	Antifungal drugs	Godeau et al. [71]
Econazole	Antifungal drugs	Firooz et al. [72]
Penconazole	Antifungal drugs	Husak et al. [73]
Cyproheptadine	Antihistamine drugs	De Bruyne et al. [74]
Felodipine	Antihypertensive drugs	Shah et al. [75]
L-Histidine	Anti-inflammatory drugs	Peterson et al. [76]
Pimozide	Antipsychotic drugs	Elmaci and Altinoz [77]
Bisacodyl	Stimulant laxative drugs	Noergaard et al. [78]
Phytosphingosine	Antimicrobial drugs	Başpınar et al. [79]
Cinchocaine	Anaesthetic drugs	Ghoniem et al. [80]
Pararosaniline	Dye agents	de Jong et al. [81]

**Table 12 molecules-27-06458-t012:** Summary of the drying methods applied to the sample leaves.

Drying Methods	Drying Process
Sun-drying	Samples were exposed to sunlight for 1 week
Shade-drying	Samples were dried in room temperature at 22–25 °C and humidity levels between 30% and 50% for 1 week
Microwave-drying	Samples were put in a microwave dryer (Samsung, Seoul, Korea) at atmospheric pressure and 160 W power (three times, 2 min each time)
Oven-drying	Samples were put in an oven (Protech, Selangor, Malaysia) at 50 °C for 24 h
Freeze-drying	Samples were frozen at −80 °C for 48 h, and then put in a freeze-dryer (Labconco, Kansas City, MO, USA) for 24 h

## Data Availability

Not applicable.

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
