# Peer review of "The Effects of Drying Techniques on Phytochemical Contents and Biological Activities on Selected Bamboo Leaves"

_molecules, 2022, doi:10.3390/molecules27196458_

Round 1
Reviewer 1 Report
The Effects of Drying Techniques on Phytochemical Contents and Biological Activities on Selected Bamboo Leaves
The study investigated the influence of the drying process of bamboo leaves on the antiradical activity and the content of polyphenols.
There is a lot of work on this remake in the literature:
· Studies on drying of imperial bamboo RE Vetter, RA Sá Ribeiro, MG Sá Ribeiro… - European Journal of …, 2015 – Springer
· Preservation and drying of bamboo W Liese, TKH Tang - Bamboo, 2015 – Springer
· A two‐stage convective air and vacuum freeze‐drying technique for bamboo shoots Y Xu, M Zhang, D Tu, J Sun, L Zhou… - … journal of food …, 2005 - Wiley Online Library
· Microwave-vacuum drying of round bamboo: A study of the physical properties HF Lv, XX Ma, B Zhang, XF Chen, XM Liu… - … and Building Materials, 2019 - Elsevier
What is the scientific novelty of this work? It is known that the best process to obtain stable preparations is lyophilization. Nothing new has been found here. Instrumental techniques were not used in the study, e.g. ordinary HPLC, which are characterized by higher precision and accuracy of determinations than spectrophotometric techniques. Perhaps then attention would be paid to other aspects of the processes under study.
The paper does not explain whether all the collected leaves had exactly the same growth and vegetation conditions? The leaves were collected randomly, and as it is known, the growing conditions have a major influence on the polyphenols formed and their type, which then may result in different directions of changes during drying, which may consequently give erroneous results after the processes. How was the representativeness of the samples ensured?
Why was an unusual wavelength used when measuring the deactivation of the DPPH radical? Typically 517 nm is used.
For what purpose was toxicity tested? Do the authors have any knowledge about the transformation of polyphenols into toxic compounds during drying? Please justify the performance of this determination in these tests.
Please adapt the literature list to the editorial requirements.
The work requires refinement and supplementation with distribution techniques.
Reviewer 2 Report
This paper is potentially interesting but there are some issues that should be carefully addressed by authors before making the paper suitable for publication in the Molecules.
The main drawback is lack of sample characterization by determining the most abundant individual phenolics using HPLC-DAD/MS.
Line 320: Please add month and year of sample collection.
Line 328: What was the amount of the sample?
Lines 339-347: Please explain why you freeze-dried infusions and then dissolved dried extract in water again.
Results and Discussion:
Did you correct results for moisture content in fresh leaves?
Line 258: You stated that ‘’ phenolic and flavonoid groups were slightly responsible for the antioxidant activities of DPPH, ABTS, and FRAP’’. Since phenolics and flavonoids are strong antioxidant, how do you explain this statement?
Round 2
Reviewer 1 Report
I recommend the article for publication.
Reviewer 2 Report
All the recommendations and issues raised have been answered and amended accordingly.